# Impact of Social Buffering and Restraint on Welfare Indicators during UK Commercial Horse Slaughter

**DOI:** 10.3390/ani13142276

**Published:** 2023-07-12

**Authors:** Katharine A. Fletcher, Georgina Limon, Barbara Padalino, Genevieve K. Hall, Natalie Chancellor, Andrew Grist, Troy J. Gibson

**Affiliations:** 1Animal Welfare Science and Ethics Group, Department of Pathobiology and Population Sciences, Royal Veterinary College, Hawkshead Lane, Hatfield AL9 7TA, UK; ghall19@rvc.ac.uk (G.K.H.); nchancellor@rvc.ac.uk (N.C.); tgibson@rvc.ac.uk (T.J.G.); 2Veterinary Epidemiology, Economics and Public Health Group, Department of Pathobiology and Population Sciences, Royal Veterinary College, Hawkshead Lane, Hatfield AL9 7TA, UK; glimon@rvc.ac.uk; 3The Pirbright Institute, Ash Road, Pirbright, Woking GU24 0NF, UK; 4Department of Agricultural and Food Sciences, University of Bologna, Viale Giuseppe Fanin 46, 40127 Bologna, Italy; barbara.padalino@unibo.it; 5Animal Welfare and Behaviour Group, School of Veterinary Sciences, University of Bristol, Langford BS40 5DU, UK; andy.grist@bristol.ac.uk

**Keywords:** slaughter, semi-feral, conspecifics, equine, restraint, handling, rifle

## Abstract

**Simple Summary:**

Retrospective CCTV footage was analysed by trained observers to assess the welfare of horses co-slaughtered with a conspecific present or slaughtered individually, restrained or unrestrained. Co-slaughtered horses were found to move around the kill pen more but were less likely to slip/fall in the pen. Both individually slaughtered horses and loose (unrestrained) horses were more likely to show agitated behaviour and resist entry to the pen, with unrestrained horses also showing increased agonistic behaviour towards abattoir personnel. Horses showed affiliative behaviour towards each other when co-slaughtered, with the shooting of the first horse seldom eliciting a startled response from the second horse. This study shows that enabling abattoirs to co-slaughter unrestrained horses could minimise stress and maximise both human safety and horse welfare. The results of this study are relevant on a wider scale, with countries across the world slaughtering high numbers of unhandled or semi-feral horses, and encourage further research to guide welfare improvements in this area.

**Abstract:**

Current legislation in the United Kingdom stipulates that horses should not be slaughtered within sight of one another. However, abattoir personnel anecdotally report that, for semi-feral horses unused to restraint, co-slaughtering alongside a conspecific could reduce distress through social buffering and improve safety, but there is a lack of evidence to support this. CCTV footage from an English abattoir was assessed retrospectively with welfare indicators from when horses entered the kill pen until they were killed. Of 256 horses analysed, 12% (32/256) were co-slaughtered (alongside a conspecific) and 88% (224/256) individually. Co-slaughtered horses moved more in the pen, but individually slaughtered horses showed more agitated behaviour, required more encouragement to enter the kill pen, and experienced more slips or falls. Unrestrained horses (40%; 102/256) showed increased agitation, movement, and agonistic behaviour towards the operator and resisted entry to the kill pen compared to restrained horses (60%; 154/256). Positive interactions between conspecifics were seen in 94% (30/32) of co-slaughtered horses, and only 6% (1/16) showed a startled response to the first horse being shot, with a median time of 15 s between shots. This study highlights the impact that both conspecific and human interactions can have on equine welfare at slaughter. Semi-feral or unrestrained horses appear to experience increased distress compared to horses more familiar with human handling, and the presence of a conspecific at slaughter mitigated this.

## 1. Introduction

An estimated 20,600 horses were slaughtered within the UK in 2021 at abattoirs and knacker yards [1]. Horse meat (*Equus caballus*) is not commonly consumed in the United Kingdom (UK). However, for horses slaughtered commercially for human food consumption in the UK, a free bullet is generally used using a 0.22 calibre long rifle. This method has been found to effectively achieve irrecoverable insensibility in horses [2], although animal and human-based factors affecting the efficacy of this method have not been explored.

Many aspects of the slaughter process can influence animal welfare—among them are human–animal interactions, with the potential for animals to experience distress due to close proximity to unfamiliar humans, as well as the novel abattoir environment. This includes handling methods, with their impacts varying depending on individual temperament, context, and the type and level of interaction. Typically, in the UK, if horses are halter-trained, they will be led into the kill pen at the abattoir to be slaughtered; however, if they are semi-feral and unable to be haltered, they will likely be herded into the kill pen. The impact of familiarity with humans and the animals’ level of domestication, in general, with regards to ease of handling, including whether they are accustomed to a halter and to restraint, has not been explored for horses at slaughter.

As a herd and prey animal, a horse observing a conspecific being slaughtered could cause distress, but research in other social species, such as pigs and sheep, found this not to be the case, especially in comparison to the impact of social isolation [3,4,5]. In some contexts, the presence of one animal has been found to mitigate a conspecific’s stress response during a stressful event, a concept referred to as social buffering [6,7]. Social buffering has been found to be predominantly influenced by the nature of the stressful stimulus presented, rather than by familiarity with the conspecific [7].

The Welfare of Animals at the Time of Killing Regulations (England) [8], which currently regulates the slaughter of animals in England, stipulates that horses should be slaughtered alone (out of sight of other horses). However, anecdotal information from slaughterhouse personnel in the UK suggests that semi-feral or unbroken horses (those that are not accustomed to wearing a halter or being handled by a human) are easier to co-slaughter, ensuring the safety of the people involved as well as reducing distress in horses. The abattoirs were previously given special permission to conduct this. There is a gap in the current scientific knowledge to evidence this either way, but the negative public perception of co-slaughter has highlighted the need for research into the positive or negative impact this might have on equine welfare at slaughter.

This study aimed to compare welfare indicators for horses that were slaughtered individually versus co-slaughtered and for horses restrained at slaughter versus unrestrained.

## 2. Materials and Methods

### 2.1. Sample

CCTV footage was obtained from a licensed abattoir in the UK which routinely slaughters horses for commercial purposes using a free-bullet rifle. With consent from the abattoir, retrospective footage was downloaded onto a secure hard drive from intermittent dates between April 2021 and August 2022. Under current English law, CCTV is required for all stages of abattoir operation where live animals are, including stunning and bleeding operations [9]. The data were provided directly by the abattoir, sequenced in chronological order by time and date, and obtained via ‘clipping’ footage of horses in the kill pen from the entry of the individual or pair of horses into the kill pen until after the horse(s) was slaughtered. The kill pen was a fully enclosed room with metal walls from floor to ceiling, approximately 5 m × 4 m (20 m^2^) in size, with entry from the lairage via a wide metal door.

Equid breed/type was recorded, but no further demographic information was able to be gathered regarding horse type, age, or origin. Whether animals appeared to be handled or unhandled (assessed through whether they were restrained when led into the kill pen using a halter and lead rope or unrestrained/loose) was also recorded. Body Condition Score was subjectively assessed through visual observation alone using a five-point scale from 1 (poor condition) to 5 (obese) [10]. Lameness was also subjectively assessed through visual observation alone and scored as not present, mildly lame, or significantly lame.

Operators were assigned an identification number (1–4) to enable differences between operators to be assessed, including the number of animals slaughtered and of which group type, alongside any associations between operator ID and welfare indicators.

### 2.2. Ethogram and Welfare Indicators

An ethogram was developed based on the literature to explore validated signs of distress or discomfort in equids [11,12,13,14,15] (Table 1). One–Zero [16] behavioural sampling was used, with behaviours recorded as present or absent. The time of observation varied, with behaviour assessed from the time the horses entered the pen until the time when they were shot.

The duration of time each horse spent in the pen, from the moment their front feet entered the pen until the time they were shot, along with the time between shooting the first horse and shooting the second horse, in instances when animals were co-slaughtered, was recorded using a digital stopwatch (Guangcailun ZSD-809).

Additional welfare indicators were recorded (Table 2), including ease that horse entered the kill pen and interactions with conspecifics (if co-slaughter or in cases where two horses were initially brought into the pen to aid entry before one was removed prior to slaughter). Human–animal interactions were recorded in the assessment, including restraint method, distance between firearm operator and horse *(estimated between terminal end of the firearm and the equine forehead)*, shot aiming position, and shot angle. Post-mortem signs of shooting effectiveness and behaviour in response to the shooting of conspecifics was also logged, along with floor condition and/or wetness, where possible. The CCTV footage was without audio, and so potential vocalisation was determined by visual observation of perceived mouth movement alone by observing whether the mouth appeared to open in a significant or prolonged way.

### 2.3. Data Collection

Primary data collection involved observation of the footage (one individual clip at a time) in chronological order by ascending date. This took place after a training session between the two observers (K.F. and G.H.), who had varying experience assessing equine behaviour, to ensure reliable use of the ethogram. K.F. and G.H. then evaluated the footage, with videos split equally between each observer, and systematically recorded equine behaviours and welfare variables.

Data were input into an Excel spreadsheet table using Microsoft Excel (version 2108).

### 2.4. Statistical Analysis

Inter-observer reliability was tested through an 8% (20/256) sample of the CCTV being analysed by both observers, and the scores were compared using Kappa coefficient analysis. Level of restraint, ear posture, potential mouth movement, shooting distance, shooting aiming position, shot angle, and lameness scored ‘Perfect’ (Kappa 1.0) agreement. Body Condition Score, ease of entry to kill pen, movement in pen, and agonistic behaviours all scored ‘almost perfect’ agreement (weighted Kappa 0.90, 0.94, 0.89, and 0.94, respectively).

All data were analysed using descriptive statistics and reported using averages and, where appropriate, medians, along with percentages for categorical variables. Chi square (or Fisher’s exact, as appropriate) tests were performed to determine if there was crude association between each behaviour variable and (i) slaughter grouping (co- versus individual) or (ii) level of restraint (unhandled/unrestrained animals vs. restrained/haltered). *p* < 0.05 was used as the indicator of significance.

Breed sub-groups were combined as either cob/draft horse type, thoroughbred/sports horse type, or native ponies [17]. Behaviour variables were grouped when deemed appropriate for further analysis. The recategorisation of variables is described in Table 3. Univariate Logistic Regression was then conducted with either slaughter grouping or level of restraint as predictor variables and each behavioural observation and outcome variable. Whether the horses were brought into the kill pen with a conspecific before one was then removed prior to slaughter was also analysed. However, these horses were included in the individual slaughter category due to low numbers (*n* = 10). Odds ratios (OR) and 95% confidence interval (CI) were calculated as measures of strength of association. Collinearity was assessed between slaughter grouping and level of restraint and between level of restraint and time in pen. These three variables exhibited strong collinearity, and therefore, univariate models were kept. SPSS (IBM SPSS Statistics 28.0.0.0, 2022) was used for all analyses.

## 3. Results

### 3.1. Descriptive Statistics

#### 3.1.1. Demographics

Footage analysed showed a total of 256 horses/ponies slaughtered (Table 4). There was a relatively equal number of both native-type ponies and thoroughbred/sports-type horses, with a smaller proportion of cob or draft-type horses. Native-type ponies constituted the majority of co-slaughtered (91%; 29/32) or unrestrained/loose (90%; 90/102) horses.

#### 3.1.2. Individual Versus Co-Slaughter

With a maximum of two horses in the kill pen at one time, 12% (32/256) of horses were co-slaughtered, and 88% (224/256) were slaughtered individually. More co-slaughtered horses showed movement while in the kill pen (62%; 20/32, compared to 33%; 73/224 of individually slaughtered horses), although more individually slaughtered horses showed excessive movement or trotting (8%; 17/224 compared to 3%; 1/32 of co-slaughtered horses). Of those horses co-slaughtered, 47% (15/32) displayed a depressed or worried body posture compared to 26% (58/224) of individually slaughtered horses, but an increased number of individually slaughtered horses showed agitated body posture (11%; 24/224; compared to 3%; 1/32 of co-slaughtered). Individually slaughtered horses required more force to enable entry to the kill pen (50%; 110/224 compared to force required for 40%; 12/32 of co-slaughtered). Additionally, no co-slaughtered horses showed aggressive behaviour to the operator or perceived mouth movement/potential vocalisation, while 3% (7/224) of individually slaughtered horses showed aggressive behaviour, and 7% (14/224) showed perceived mouth movement. No co-slaughtered horses were seen to slip or fall, while 13% (30/224) of individually slaughtered horses were seen to slip (Table 5).

Of those co-slaughtered, 94% (30/32) showed positive interactions and affiliative behaviour between conspecifics, and only one horse (3%) that was co-slaughtered showed negative behaviour towards a conspecific. When analysing the animal’s behaviour in response to the shooting of a conspecific, 94% (15/16) showed no fear behaviour/nothing of note, and 6% (1/16) showed startled/fearful behaviour.

#### 3.1.3. Restraint

The majority of restrained horses (those wearing a halter and lead rope) were shot individually (98%; 151/153), with just three horses co-slaughtered wearing a halter. Of those horses slaughtered unrestrained (40%; 103/256), the majority were shot at a distance of >2 m (84%; 87/103), with 31% (27/87) of those co-slaughtered and 69% (60/87) individually. Just 3% (3/103) of unrestrained horses were shot at point-blank range, with one of these being in a pair and two being individually shot. Only one horse that was co-slaughtered unrestrained showed agitated body posture, and only one horse from that cohort showed significant movement in the pen (Table 6).

#### 3.1.4. Operator Details

There were four operators responsible for the shooting in the footage analysed (Table 7). Operator A was the main operator, slaughtering 70% (179/256) of the horses sampled. Operator A was also the only operator to co-slaughter horses, of which 6% (2/32) were haltered/restrained, and 94% (30/32) were unrestrained/loose.

#### 3.1.5. Details of Shooting

All horses demonstrated behavioural signs associated with an effective shot, including immediate collapse and/or lack of righting reflex and leg kicking/spasms or rigidity. One co-slaughtered horse and two individually slaughtered horses needed two shots (both of which were unrestrained and shot from >1 m), with the remainder (98%; 252/256) shot successfully first time. The majority of horses were shot at an angle of between >45 to <90 degrees (82%; 193/235), with 18% (42/235) of horses shot at <45 degrees and only one horse assessed as being shot at an angle of >90 degrees. A small proportion of horses (8%; 20/256) were unable to be assessed for shot angle due to this being out of view on the footage.

#### 3.1.6. Time in Kill Pen

The median time for horses to spend in the kill pen was 21 s (range: 4–250 s), with the median time between shots being 15 s (range: 4–58 s). Co-slaughtered horses spent longer in the kill pen, as did unrestrained horses (Figure 1).

### 3.2. Logistic Regression

#### 3.2.1. Details of Shooting

There was no significant difference in aiming position between co- or individually slaughtered or restrained or unrestrained cohorts. However, co-slaughtered horses were more likely to be shot at an angle of more than 90 degrees (*p* = 0.003, OR: 31.80, 95% CI, 3.16–320.09) and more likely to be shot at a distance (>15 cm) (*p* < 0.001, OR: 28.08, 95% CI, 6.54–120.60). Unrestrained animals were also more likely to be shot at a distance (>15 cm) (*p* < 0.001, OR: 912.50, 95% CI, 189.80–4387.10), with the operator commonly holding the halter of restrained animals prior to shooting, resulting in a closer shot.

#### 3.2.2. Co- versus Individual Slaughter

There was a significant difference in slipping/falling (*p* = 0.03), with no co-slaughtered horses seen to slip or fall. A significant difference was found between horses slaughtered individually or co-slaughtered for movement in pen (*p* = 0.002), with co-slaughtered horses 3.45 times more likely to show movement in the pen (95% CI, 1.6–7.43) (Table 8). Slips/falls were found to be more significant (*p* = 0.03), but perceived mouth movement was not significant (*p* = 0.35). However, a logistic regression model was not run on these categories due to >1 category having a count of zero observations.

#### 3.2.3. Interactions with Conspecifics

Of those slaughtered individually, 10 were led in with a conspecific before one horse was removed for the remaining horse to be slaughtered (Table 8). This happened where one of the pair was unrestrained, so presumably was to assist with entry to the pen, although there was no significant association between ease of entry and whether a horse was led in with a companion. There was a significant association between those led in with a companion and type of horse, with those led in with a companion more likely to be native-type ponies (*p* < 0.001, OR: 0.43, 95% CI, 0.29–0.64). There was no significant association between slips or falls (*p* = 0.24), ear posture, behaviour towards operator, or perceived mouth movement, but being led in with a conspecific did appear to reduce movement in the pen (*p* < 0.001, OR 4.59, 95% CI, 2.27–9.28) and reactive body posture (*p* = 0.02, OR: 2.25, 95% CI, 1.15–4.38).

#### 3.2.4. Restraint

There was a significant association in restraint (*p* = 0.001) between slaughter grouping and level of restraint (unrestrained vs. halter). When restraint was analysed as a separate factor to slaughter grouping, there was a significant difference in movement in the pen (*p* < 0.001) between horses slaughtered restrained or unrestrained, with unrestrained horses more likely to show movement (Table 8). Unrestrained horses were also more likely to show reactive body posture. Restraint also appeared to effect ear posture, ease of entry, and behaviour to operator, with unrestrained horses more likely to show backwards ears, require force to enter the kill pen, and show agonistic behaviour to the operator. There was no significant association between restraint and possible vocalisation (*p* = 0.41). There was also a significant association between slips and falls and level of restraint (*p* = 0.04). However, a logistic regression model was not run on these categories due to >1 category having a count of zero observations.

## 4. Discussion

This study found that the presence or absence of a conspecific for horses at slaughter, contrary to anecdotal or public perception, was unlikely to significantly affect the expression of behaviour. However, the results of this study showed a trend towards increased willingness to enter the kill pen, reduced agitation, and a reduced likelihood of slipping or falling when a conspecific was present during slaughter. Additionally, there was a significant difference in behaviour between horses that were restrained and unrestrained at slaughter, suggesting that an animal’s experience with previous handling and their ability to be restrained or haltered could impact their welfare at slaughter.

It can be difficult to evaluate whether a horse is halter-trained or ‘broken’ and used to handling or not, with official veterinarians or animal welfare officers at an abattoir not having a standardised test to determine this. Council Regulation (EC) No. 1/2005 [18] on the protection of animals during transport requires that, for transportation, unbroken animals must travel in groups rather than individually, with an unbroken horse defined in this Regulation as a horse that “cannot be tied or led by a halter without causing avoidable excitement, pain or suffering” [18]. However, there are no such regulations for grouping them or separating them when they are slaughtered, beyond the UK legal requirement that equids should be stunned/killed individually, regardless of their behaviour or experience with handling [8]. It is also important to consider that individual horses will be affected by previous experience with humans [19,20] and by different stressors or contexts, but there is an absence of studies investigating these factors at slaughter.

A test to determine whether a horse is broken or unbroken (familiar with handling or unfamiliar) by assessing avoidance behaviour during handling has been developed and validated for use in transport and farm contexts, including on arrival at the abattoir [13]. Unhandled or ‘unbroken’ horses were seen to display increased avoidance behaviours, flight reactions, and significantly longer displacement behaviours [21]. Such a test would be helpful to incorporate into a more formal assessment of horses at slaughter but arguably could cause increased stress at a time when emotions are already heightened, and with the present study using retrospective CCTV footage, the test would not have been feasible in these circumstances.

Although there are limitations in using CCTV footage rather than in-person analysis, preventing the ability to assess more detailed signs of insensibility—e.g., eye reflexes, post-shot, or more subtle behavioural signs such as facial expression—or evaluate any other circumstances that may have occurred during lairage, the use of CCTV footage ensures that the study is unbiased by any observer effect. It also enables footage to be reviewed multiple times to enable a high level of accuracy regarding timing and ethogram assessment. The unbalanced sample size for this study, with the small number of co-slaughtered horses and only one operator co-slaughtering horses, are potential limitations, particularly regarding statistical analysis. However, ensuring equal sample size was not possible due to the operational nature of the study and the inability to obtain footage prior to April 2021.

The high observation of positive interactions recorded between conspecifics during co-slaughter and lack of perceived mouth movement, which could indicate possible vocalisation for co-slaughtered horses, cautiously suggests the presence of social buffering during slaughter. This demonstrates that, regardless of familiarity, equids have a desire to seek each other and engage in body contact in the abattoir and provide mutual support through affiliative behaviour. This corresponds with research into other livestock at slaughter, where cattle have been found to express motivation to stay with conspecifics during slaughter and showed a negative reaction to social separation [22]. Similarly, in sheep, isolation from conspecifics during slaughter has been found to increase associated stress hormones [23]. In the present study, the fear response shown by some co-slaughtered horses to the first animal being shot was observed less frequently than expected, but with a short time between shots and only two horses in the pen at any time, this could arguably cause less prolonged or intense distress than the separating of conspecifics for the purposes of slaughter. However, being brought into the pen with a conspecific, which was then removed prior to slaughter, did not seem to be enough to mitigate stress behaviour.

Stress behaviour presented itself in unrestrained horses through an increase in backwards ear posture, reactive body posture, movement in the pen, and agonistic behaviour towards the operator. Although some movement could arguably suggest a better welfare state than a motionless, potentially apathetic animal, or one in a freeze state, it could also represent increased distress when compared to an animal standing relaxed. Minimising movement in the pen and reactive body posture could potentially have the additional consequence of improving shooting performance. Considering only three horses were seen receiving a second shot, this suggests a high success rate and skilled firearms operators in comparison to other methods of slaughter, such as a penetrative captive bolt, where inadequate stunning occurs more often [24]. A captive bolt also requires adequate restraint to ensure that the horse can be shot at point-blank range [25], while previous research has found that, for a free-bullet rifle, regardless of whether the horse was shot at point-blank range or at a distance of approximately two metres, animals showed near immediately irrecoverable loss of consciousness, associated with gross macroscopic damage to the brainstem and/or cerebral lobes in all cases [2]. In the present study, shot angle and distance did not appear to be associated with shooting effectiveness based on visual observation of signs of consciousness.

Due to the retrospective nature of the study, it was not known whether animals in poorer body condition or showing signs of lameness were prioritised for slaughter. The prevalence of avoidance behaviour towards abattoir personnel observed during co-slaughter suggests that co-slaughtered horses were semi-feral or unaccustomed to human contact, although this could not be confirmed by the abattoir and no co-slaughtered horses displayed aggressive behaviour.

Native ponies appeared to be more likely to be co-slaughtered, potentially because these types of ponies are more likely to be semi-feral or unrestrained, or this could be part of the abattoir’s decision regarding operator safety and ease of handling. The bias towards this breed type and associated temperament for co-slaughter could also have influenced behaviour towards the operator and propensity towards aggression or avoidance. Assumptions upon origin and previous handling experience were made based on the type of horse and their behaviour, without being able to link results to passports or the history of each horse. Further research to empirically assess what previous handling horses at slaughter have received and how this impacts their behaviour and welfare at slaughter would be valuable. Previous studies have shown that, in other countries, high numbers of feral horses are sent for slaughter, with these horses not accustomed to or familiar with being haltered prior to arrival at the abattoir [26,27], including in countries where they are specifically bred for meat production and receive minimal handling prior to slaughter [27]. This highlights the importance and relevance of research in this area and of the present study’s findings to enable the welfare of all types of horses sent for slaughter to be safeguarded, not only in the United Kingdom but in other countries where horse slaughter is more common.

## 5. Conclusions

This study was the first of its kind to empirically analyse the impact that the presence of a conspecific can have on equine behaviour and welfare at slaughter. Semi-feral or unhandled horses were found to experience increased stress, and with large numbers of these horses being sent to slaughter across the world, these findings could have substantial repercussions for the global horse meat production industry. The presence of a conspecific at slaughter for these types of horses to potentially prompt social buffering and mitigate severe stress is an option that should be further explored. This could also result in a potentially diminished safety risk to horses and humans, enhance the speed of the processing line, and consequently improve the welfare of horses at slaughter.

## Figures and Tables

**Figure 1 animals-13-02276-f001:**
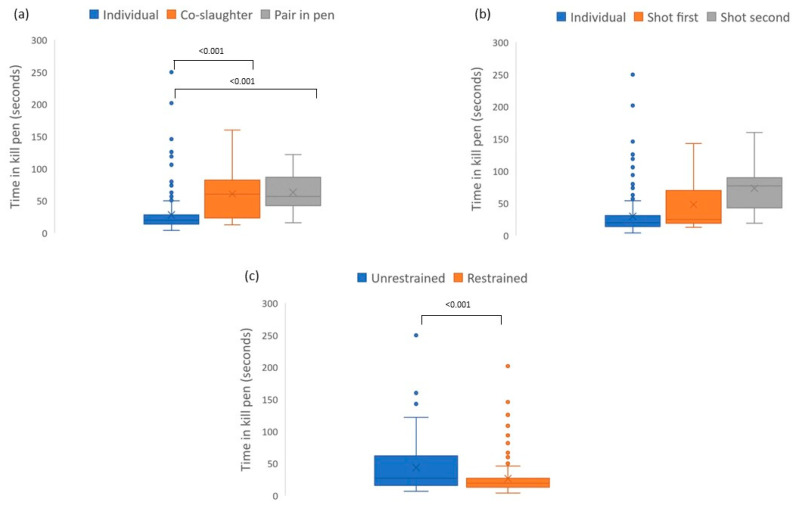
Relationship between time spent in kill pen prior to slaughter. (**a**) Whether the animal was in the pen individually, brought in with a conspecific and co-slaughtered, or brought in with a conspecific, which is then removed, leaving the animal alone for shooting. (**b**) Type of grouping in pen and shot order. (**c**) Restraint method, showing differences in time for all these factors (slaughter grouping, shot order, and level of restraint).

**Table 1 animals-13-02276-t001:** Ethogram used to assess equine behaviours in the kill pen, adapted from previously published literature on equine behaviour.

Behavioural Categories	Behaviour	Description of Behaviour	Reference
Body posture when in kill pen	Standing quietly/calm	Equine stood up on all four limbs, immobile	Regan et al. (2014) [11]
	Startled	Tense posture, head held erect and high. Often characterised by a jump and/or movement	Torcivia & McDonnell, (2021) [12]
	Restless/agitated	Excessive movement and irritability	Torcivia & McDonnell, (2021) [12]
	Depressed	Little movement. Standing up but head low to floor	Torcivia & McDonnell, (2021) [12]
	Alert	Stance is rigid. Neck elevated but not vertical, and head orientated towards operator. Ears forwards	Torcivia & McDonnell, (2021) [12]
Agonistic behaviours to operator	Avoidance/ retreat	Moves or attempts to move away/turns head away	Burn et al. (2010) [13]
	Aggressive/rear	Both front limbs lifted off the floor. Neck elevated to a vertical position	McDonnell & Haviland, (1995) [14]
	Aggressive/bite/bite threat	Rapid opening and closing of jaw	McDonnell & Haviland, (1995) [14]
	Aggressive/kick affiliative	One or both hind legs extended out rapidly. With or without an aim Turns head towards operator/ears forward	McDonnell & Haviland, (1995) Burn et al. (2010) [14]
Ear posture at time of shot	Forwards	One or both ears facing forwards, ear cups fully visible if facing the equine head-on	Regan et al. (2014) [11]
	Backwards	Both ears backwards, ear cups visible if standing behind the equine	Regan et al. (2014) [11]
	To the side	Both ears sideways, ear cups orientated at 90 degrees lateral to forward facing position	Regan et al. (2014) [11]
Locomotory behaviour	Slips Falls Nothing of note	A loss of balance, without any part of the body (other than hooves) touching the ground A loss of balance, causing any part of the body (other than hooves) to touch the ground (observed in corridor to kill pen/the kill pen)	Felici et al. (2022) [15]

**Table 2 animals-13-02276-t002:** Welfare indicators recorded when examining footage, including indicators relating to both conspecific and human–horse interactions, and indicators relating to environment.

Measurement	Categories
Level of restraint	0 = No restraint—feral/no halter; 1 = Restraint used—halter and lead, but no/minimal pressure applied; 2 = Moderate restraint—pressure applied to halter; 3 = High restraint—tight halter and pressure.
Movement in kill pen	0 = Minimal movement/stationary; 1 = Some movement/walking; 2 = Excessive movement/trotting.
Shot aiming position	Frontal; Lateral; Other
Shot angle	Calculated subjectively as observed on footage as <45°, <90°, or >90°
Shot effectiveness	Signs of insensibility/unconsciousness: defined by immediate collapse + lack of righting reflex + leg kicking or muscle spasms or rigidity
Number of shots	As observed on footage by whether operator appears to point gun a second time after initial firing
Behaviour in response to (immediately after) the shooting of conspecifics	Startled/fearful; No response.
Perceived mouth movement/ potential vocalisation	Mouth seen to open significantly and prolonged, with tension in jaw
Distance between firearm and horse	1 = Point blank (muzzle/forehead)—direct contact shot; 2 = 15 cm–1 m; 3 = From afar (>1 m).
Ease that horse enters the kill pen	1 = Willingly, without pressure/force; 2 = Encouragement/gentle pressure required; 3 = Tension/force required.
Interactions between conspecifics	Positive (e.g., mutual grooming, affiliative behaviour, body contact); Negative (bite/kick, threat to bite/kick, and/or avoidance); None of note
Body condition	Five-point scale: 1 = Poor; 2 = Moderate; 3 = Good; 4 = Fat; 5 = Obese (Carroll & Huntington, 1988) [10]
Floor hazards	Obstructions on the floor: Present; Absent
Floor wetness	Acceptable (clean, minimal water); Unacceptable (significant water/blood/faeces observed)

**Table 3 animals-13-02276-t003:** Recategorisation of behavioural variables to enable binary logistic regression analysis.

Variable	Original Categories	Recategorisation into Binary Categories for Analysis
Movement in pen	No movement/stationary Some movement/walking Excessive Movement/trotting	Stationary Movement
Body posture in pen	Calm Depressed/worried Agitated	Non-reactive Reactive
Ear posture	Forwards To the side Backwards	Forwards/side Backwards
Ease of entry	Willingly/without force Mild pressure/force required Considerable force/pressure	Without force With force
Behaviour to operator	Affiliative Avoidant Aggressive	Affiliative Agonistic

**Table 4 animals-13-02276-t004:** Descriptive parameters regarding breed type of horse, Body Condition Score, and perceived lameness recorded when examining footage.

Descriptor	Population Number (%) Total: 256	Co-slaughtered Number (%) Total: 32	Individual Number (%) Total: 224	Restrained Number (%) Total: 154	Unrestrained Number (%) Total: 102
Breed type					
- Native;	116 (45%)	29 (91%)	87 (39%)	24 (16%)	92 (90%)
- Cob/Draft;	23 (9%)	2 (6%)	21 (9%)	21 (14%)	2 (2%)
- Thoroughbred (TB)/Sports.	117 (46%)	1 (3%)	116 (52%)	109 (70%)	8 (8%)
Body Condition Score					
*(Carroll & Huntington, 1988)* [10]					
- 1 (poor);	5 (2%)	0 (0%)	5 (2%)	4 (3%)	1 (1%)
- 2 (moderate);	36 (14%)	2 (6%)	33 (15%)	29 (19%)	6 (6%)
- 3 (good);	199 (78%)	26 (81%)	174 (78%)	116 (75%)	84 (82%)
- 4 (fat);	16 (6%)	4 (13%)	12 (5%)	5 (3%)	11 (11%)
- 5 (obese).	0 (0%)	0 (0%)	0 (0%)	0 (0%)	0 (0%)
Lameness					
- None;	238 (93%)	31 (97%)	207 (92%)	139 (90%)	100 (98%)
- Mild;	15 (6%)	1 (2%)	14 (6%)	13 (9%)	1 (1%)
- Significant.	3 (1%)	0 (0%)	3 (2%)	2 (1%)	1 (1%)

**Table 5 animals-13-02276-t005:** Behaviours seen in horses, stratified by slaughtered in pairs or individually; according to order of shooting (first vs. second shot); and restrained (haltered) or unrestrained (loose), including data collected on ease of entry into the kill pen, movement in the pen, body and ear posture, slips or falls, perceived vocalisation, and behaviour towards operator.

Variable	Slaughtered Individually Number (%)	Slaughtered in Pair Number (%)	Shot First Number (%)	Shot Second Number (%)	Unrestrained Number (%)	Restrained Number (%)
Pre-slaughter handling						
- Individual;	224 (88%)	-	-	-	29 (28%)	151 (98%)
- Paired.	-	32 (12%)	16 (50%)	16 (50%)	73 (72%)	3 (2%)
Movement in pen						
- No movement/stationary;	151 (67%)	12 (38%)	10 (63%)	2 (13%)	45 (45%)	118 (77%)
- Some movement;	56 (25%)	19 (59%)	6 (37%)	13 (81%)	49 (48%)	26 (18%)
- Significant Movement;	17 (8%)	1 (3%)	0 (%)	1 (6%)	8 (7%)	10 (6%)
- Missing data.	0	0	0	0	0	0
Body posture in pen						
- Calm;	142 (63%)	16 (50%)	10 (63%)	6 (37%)	39 (38%)	119 (77%)
- Depressed/worried;	58 (26%)	15 (47%)	5 (31%)	10 (63%)	56 (55%)	17 (11%)
- Agitated;	24 (11%)	1 (3%)	1 (6%)	0 (0)	5 (5%)	18 (12%)
- Missing data.	0	0	0	0	0	0
Slips/falls						
- Absent;	194 (87%)	32 (100%)	16 (50%)	16 (50%)	85 (83%)	141(92%)
- Present;	30 (13%)	0 (0%)	0 (0%)	0 (0%)	17 (17%)	13 (8%)
- Missing data.	0	0	0	0	0	0
Ear posture						
- Forwards;	98 (48%)	10 (43%)	9 (56%)	8 (50%)	54 (69%)	54 (37%)
- To the side;	66 (33%)	9 (39%)	7 (44%)	8 (50%)	17 (22%)	58 (39%)
- Backwards;	39 (19%)	4 (17%)	0 (0%)	0 (0%)	7 (9%)	36 (24%)
- Missing data.	21	9	0	0	24	6
Perceived mouth movement						
- Absent;	198 (93%)	30 (100%)	16 (50%)	16 (50%)	84 (93%)	144 (95%)
- Present;	14 (7%)	0 (0%)	0 (0%)	0 (0%)	6 (7%)	8 (5%)
- Missing data.	11	2	0	0	12	2
Ease of entry						
- Willingly/without force;	112 (50%)	18 (60%)	9 (60%)	9 (60%)	21 (21%)	109 (72%)
- Mild pressure/force;	55 (25%)	8 (27%)	4 (27%)	4 (27%)	37 (37%)	26 (17%)
- Considerable force;	55 (25%)	4 (13%)	2 (13%)	2 (13%)	42 (42%)	17 (11%)
- Missing data.	2	2	1	1	2	2
Behaviour to operator						
- Affiliative;	101 (45%)	17 (53%)	9 (56%)	8 (50%)	23 (23%)	95 (62%)
- Avoidant;	116 (52%)	15 (47%)	7 (44%)	8 (50%)	79 (77%)	52 (34%)
- Aggressive;	7 (3%)	0 (0%)	0 (0%)	0 (0%)	0 (0%)	7 (4%)
- Missing data.	0	0	0 (0%)	0 (0%)	0	0

**Table 6 animals-13-02276-t006:** Behaviours seen in horses slaughtered unrestrained, individually or in pairs.

Variable	Slaughtered Individually Number (%) Total: 71% (*n* = 73)	Slaughtered in Pair Number (%) Total: 29% (*n* = 30)
Movement in pen		
- No movement/stationary;	36 (49%)	9 (30%)
- Some movement;	30 (41%)	20 (67%)
- Significant movement;	7 (10%)	1 (3%)
- Missing data.	0	0
Body posture in pen		
- Calm;	25 (34%)	15 (50%)
- Depressed/worried;	42 (58%)	14 (47%)
- Agitated/startled;	6 (8%)	1 (3%)
- Missing data.	0	0
Slips/falls		
- Present;	17 (13%)	0 (0%)
- Absent.	56 (77%)	30 (100%)
Ear posture		
- Forwards;	45 (78%)	10 (48%)
- To the side;	10 (17%)	7 (33%)
- Backwards;	3 (5%)	4 (19%)
- Missing data.	15	9
Mouth movement		
- Absent;	52 (85%)	30 (100%)
- Present;	9 (15%)	0 (0%)
- Missing data.	12	0
Ease of entry		
- Willingly/without force;	4 (5%)	18 (64%)
- Mild force required;	29 (40%)	8 (29%)
- Significant force required;	40 (55%)	2 (7%)
- Missing data.	0	2
Behaviour to operator		
- Affiliative;	9 (12%)	15 (50%)
- Avoidant;	64 (88%)	15 (50%)
- Aggressive.	0 (%)	0 (%)

**Table 7 animals-13-02276-t007:** Behaviours seen in horses slaughtered and shooting details, comparing those slaughtered restrained (haltered) vs. slaughtered unrestrained (loose).

Operator ID	A Number (%) Total: 70% (*n* = 179)	B Number (%) Total: 9% (*n* = 23)	C Number (%) Total: 19% (*n* = 49)	D Number (%) Total: 2% (*n* = 5)
Pre-slaughter handling				
- Individual;	147 (82%)	23 (100%)	49 (100%)	5 (100%)
- Co-slaughter;	32 (18%)	0 (0%)	0 (0%)	0 (0%)
- Haltered;	93 (52%)	18 (78%)	37 (76%)	5 (100%)
- Unhaltered/loose.	86 (48%)	5 (22%)	12 (24%)	0 (0%)
Movement in pen				
- No movement/stationary;	101 (56%)	16 (70%)	42 (86%)	4 (80%)
- Some movement/walking;	62 (35%)	6 (26%)	6 (12%)	1 (20%)
- Excessive movement/trotting;	16 (9%)	1 (4%)	1 (2%)	0 (0%)
- Missing data.	0	0	0	0
Body posture in pen				
- Calm;	107 (60%)	17 (74%)	30 (61%)	4 (80%)
- Depressed/worried;	55 (31%)	1 (4%)	17 (35%)	0 (0%)
- Agitated/startled;	17 (9%)	5 (22%)	2 (4%)	1 (20%)
- Missing data.	0	0	0	0
Slips/falls				
- Absent;	155 (87%)	21 (91%)	45 (92%)	5 (100%)
- Present;	24 (13%)	2 (9%)	4 (8%)	0 (0%)
- Missing data.	0	0	0	0
Ear posture				
- Forwards;	69 (45%)	11 (52%)	26 (57%)	2 (40%)
- To the side;	56 (36%)	3 (14%)	15 (33%)	1 (20%)
- Backwards;	29 (19%)	7 (33%)	5 (11%)	2 (40%)
- Missing data.	25	2	3	0
Perceived mouth movement				
- Absent;	158 (92%)	21 (100%)	44 (98%)	5 (100%)
- Present	13 (8%)	0 (0%)	1 (2%)	0 (0%)
- Missing data.	8	2	4	0
Ease of entry				
- Willingly/without force;	83 (47%)	11 (48%)	32 (65%)	4 (80%)
- Mild force required;	48 (27%)	7 (30%)	8 (16%)	0 (0%)
- Significant force required;	44 (25%)	5 (22%)	9 (18%)	1 (20%)
- Missing data.	4	0	0	0
Behaviour to operator				
- Affiliative;	85 (47%)	3 (13%)	26 (53%)	4 (%)
- Avoidant;	88 (49%)	19 (83%)	23 (47%)	1 (%)
- Aggressive;	6 (4%)	1 (4%)	0 (%)	0 (%)
- Missing data.	0	0	0	0

**Table 8 animals-13-02276-t008:** Behaviours seen in horses slaughtered, comparing those slaughtered individually vs. co-slaughtered; those slaughtered to the kill pen with a conspecific vs. slaughtered alone; and those slaughtered restrained (haltered) vs. slaughtered unrestrained (loose) (significant *p*-values in bold). *Ref* = reference category for outcome.

Variable	Movement (Stationary vs. Movement)		Body Posture (Reactive vs. Non-reactive)		Ear Posture (Forwards/Side vs. Backwards)		Ease of Entry (No Force vs. with Force)		Behaviour to Operator (Affiliative vs. Agonistic)	
Factor (n)	OR (95% CI)	*p*-Value	OR (95% CI)	*p*-Value	OR (95% CI)	*p*-Value	OR (95% CI)	*p*-Value	OR (95% CI)	*p*-Value
Slaughter type										
- Individual (224);	*Ref*		*Ref*		*Ref*		*Ref*		*Ref*	
- Co-slaughtered (32).	3.45 (1.6–7.43)	**0.002**	1.73 (0.82–3.65)	0.14	1.11 (0.50–2.44)	0.81	1.48 (0.68–3.20)	0.31	1.38 (0.66–2.90)	0.4
Entry										
- With conspecific then removed (42);	*Ref*		*Ref*		*Ref*		*Ref*		*Ref*	
- Brought in alone (214).	4.59 (2.27–9.28)	**<0.001**	2.25 (1.15–4.38)	**0.02**	1.23 (0.61–2.48)	0.56	1.22 (0.62–2.39)	0.57	0.93 (0.48–1.80)	0.83
Restraint										
- Haltered (153);	*Ref*		*Ref*		*Ref*		*Ref*		*Ref*	
- Unhaltered/loose (103).	4.16 (2.42–7.13)	**<0.001**	5.50 (3.17–9.51)	**<0.001**	3.80 (2.15–6.71)	**<0.001**	9.54 (5.25–17.32)	**<0.001**	5.53 (3.14–9.75)	**<0.001**

## Data Availability

Restrictions apply to the availability of these data. Data were obtained from the participating abattoir, with appropriate informed consent, and are available from the authors with the permission of the abattoir.

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
