# Peer review of "Impact of Social Buffering and Restraint on Welfare Indicators during UK Commercial Horse Slaughter"

_animals, 2023, doi:10.3390/ani13142276_

Round 1
Reviewer 1 Report
I have also included the below comments as a word file:
Overall comment:
This is an interesting article that puts a spotlight on slaughter, an important area of welfare. Results appear to be mixed and some clarifications are needed.
I have made a few suggestions below:
Introduction
L50 ‘in the course of business’ is a little unclear. Are you referring to food consumption here even though it is not common?
L66 Remove ‘however’ as this will link your two points better
L76 Is pair slaughtering widely practiced then, despite the fact that it doesn’t conform to the current regulations? Your study abattoir obviously practices pair slaughtering and is UK based.
L77 I would add here whether the public opinion is generally positive or negative as many readers won’t be aware.
Materials and Methods
Table 2: Some increased clarity is needed for level of restraint – if 1 is use of a halter do categories 2 and 3 refer to different types of restraint or to the level of force used with the restraint and how was this quantified?
L157: This is the first time that bringing in animals in pairs but one animal leaving has been mentioned – I think this needs some explanation or introduction potentially earlier in the paper. You will also need to explain how this affected your results/groupings as this is somewhat of a middle category between individual and pair slaughter (some buffering could take place despite the horse being alone at the actual point of slaughter?)
Table 3: Please change the heading ‘categories combined for analysis’ as this suggests that the two categories listed in this column were combined which I don’t believe is the case – if I understand correctly these are the binary categories left which you analysed.
Results
L180-183 “Paired-slaughtered horses moved more whilst in the kill pen (62%; 20/32, compared to 33%; 73/224 of individually slaughtered horses), although more individually slaughtered horses showed significant movement (8%; 17/224 compared to 3%; 1/32 of paired slaughtered horses)” How does this relate to your table 2 categories of minimal movement, walking, excessive movement/trot? Try to keep the terminology consistent throughout. Also, what is some vs significant movement (in table 5), how is this determined and how is this viewed in relation to welfare? For example, some movement could be a better welfare state than an apathetic animal but it could also represent a distressed state when compared to a relaxed standing animal.
Tables 5 and 6 are similar in content, is there a way that all variables could be included in one table, or one table could be put in supplementary material?
In terms of behaviour towards operator on the premise of being pair slaughtered, is there not the potential here for aggression or lack of it to be breed temperament related as the majority of pair slaughtered animals were native ponies vs individually slaughtered thoroughbreds? Perhaps something to bring up in the discussion?
Table 7 has percentages missing in the top of column A
L228 You state here that 3 horses needed two shots, in L346 you state two, please amend.
L237 Amend to horses
For Fig 1a. Please clarify there difference here between co-slaughter and pair in pen. Is your graph a and b here the same as in your key – I think they may be switched around?? It may also be useful throughout to check on the terminology used and clarify if co-slaughter and pair slaughter are used interchangeably.
Discussion
L292 Amend to results
L325 The limitations need to be expanded to acknowledge the small sample size of pair slaughtered animals, the bias in breed for this type of slaughter and the fact that only one operator performed pair slaughter.
L337-339 I think this is a fair point in the format your abattoir operated (with only one other horse in the kill pen and it being shot very soon after the first). Potentially caution would need to be taken with operationalising this to ensure that it is not interpreted as ‘horses aren’t distressed when seeing another horse killed’ so license is granted to have many animals present and/or long waits before the ‘witnessing animal’ is slaughtered.
L371 Square brackets needed
Conclusion
377-379: I think more caution is needed here in your interpretation as you did not directly measure the handling status and familiarity of the animals with humans prior to their slaughter. Perhaps focus more on the paired vs individual aspect. Although I agree that the study results are important to take into account in countries where larger amounts of feral horses are slaughtered.
Author Response
On behalf of my co-authors and myself, I would like to express our sincere gratitude to the reviewers for their useful feedback regarding our research article, entitled ‘Impact of social buffering and restraint on welfare indicators during UK commercial horse slaughter’. I have responded to the reviewers’ comments both online and, for your convenience, below, and attach a revised version of the article.
Reviewer 1:
Overall comment:
This is an interesting article that puts a spotlight on slaughter, an important area of welfare. Results appear to be mixed and some clarifications are needed.
I have made a few suggestions below:
Introduction
L50 ‘in the course of business’ is a little unclear. Are you referring to food consumption here even though it is not common?
Yes, referring to food consumption, have amended to read “food consumption” instead of “human” and deleted the “i.e. in the course of business”.
L66 Remove ‘however’ as this will link your two points better
Addressed
L76 Is pair slaughtering widely practiced then, despite the fact that it doesn’t conform to the current regulations? Your study abattoir obviously practices pair slaughtering and is UK based.
It was permitted under special circumstances, prior to an increase in public awareness and negative backlash causing it to be stopped. Have added: “The abattoir were previously given special permission to conduct this.”
L77 I would add here whether the public opinion is generally positive or negative as many readers won’t be aware.
Addressed to clarify public opinion is generally negative.
Materials and Methods
Table 2: Some increased clarity is needed for level of restraint – if 1 is use of a halter do categories 2 and 3 refer to different types of restraint or to the level of force used with the restraint and how was this quantified?
Have edited it to improve clarity to 1= restraint used – halter & lead, but no/minimal pressure applied
2= moderate restraint/pressure applied to halter
L157: This is the first time that bringing in animals in pairs but one animal leaving has been mentioned – I think this needs some explanation or introduction potentially earlier in the paper. You will also need to explain how this affected your results/groupings as this is somewhat of a middle category between individual and pair slaughter (some buffering could take place despite the horse being alone at the actual point of slaughter?)
Good point, we looked into whether there was an impact on stress behaviour of horses brought in as a pair before one was removed versus horses brought individually. I have clarified this and also mentioned it on line 125: “interactions with conspecifics (if co-slaughter or in cases where two horses were initially brought into the pen to aid entry before one was removed prior to slaughter).”
Table 3: Please change the heading ‘categories combined for analysis’ as this suggests that the two categories listed in this column were combined which I don’t believe is the case – if I understand correctly these are the binary categories left which you analysed.
Have changed to ‘recategorisation into binary categories for analysis’. Apologies for the confusion.
Results
L180-183 “Paired-slaughtered horses moved more whilst in the kill pen (62%; 20/32, compared to 33%; 73/224 of individually slaughtered horses), although more individually slaughtered horses showed significant movement (8%; 17/224 compared to 3%; 1/32 of paired slaughtered horses)” How does this relate to your table 2 categories of minimal movement, walking, excessive movement/trot? Try to keep the terminology consistent throughout. Also, what is some vs significant movement (in table 5), how is this determined and how is this viewed in relation to welfare? For example, some movement could be a better welfare state than an apathetic animal but it could also represent a distressed state when compared to a relaxed standing animal.
Apologies for the oversight in using inconsistent terminology, have altered so that it is consistent, and added a sentence in the discussion on line 365 regarding how movement might not necessarily indicate welfare compromise.
Tables 5 and 6 are similar in content, is there a way that all variables could be included in one table, or one table could be put in supplementary material?
I have kept Table 6 separate because it is a subset table so would be difficult to combine with Table 5 as it could be misleading in terms of sample size. I feel it is important to include as it is explaining how both factors (individual versus co and restrained versus unrestrained) impact on one another to show how co-slaughter versus individual specifically impacts most significantly on unrestrained horses.
In terms of behaviour towards operator on the premise of being pair slaughtered, is there not the potential here for aggression or lack of it to be breed temperament related as the majority of pair slaughtered animals were native ponies vs individually slaughtered thoroughbreds? Perhaps something to bring up in the discussion?
Yes good point, now included in the discussion in line 389: “The bias towards this breed type – and associated temperament – for co-slaughter could also have influenced behaviour towards the operator and propensity towards aggression or avoidance.”
Table 7 has percentages missing in the top of column A
Addressed.
L228 You state here that 3 horses needed two shots, in L346 you state two, please amend.
Addressed – L346 changed to three.
L237 Amend to horses
Addressed.
For Fig 1a. Please clarify there difference here between co-slaughter and pair in pen. Is your graph a and b here the same as in your key – I think they may be switched around?? It may also be useful throughout to check on the terminology used and clarify if co-slaughter and pair slaughter are used interchangeably.
Amended key/caption and changed description to ‘Time spent in kill pen prior to slaughter and (a) whether the animal was in the pen individually; brought in with a conspecific and co-slaughtered; or brought in with a conspecific, which is then removed leaving the animal alone for shooting, (b) type of grouping in pen and shot order, and (c) restraint method, showing differences in time for all these factors (slaughter grouping, shot order and level of restraint).’ Have also ensured terminology is used consistently throughout.
Discussion
L292 Amend to results
Addressed.
L325 The limitations need to be expanded to acknowledge the small sample size of pair slaughtered animals, the bias in breed for this type of slaughter and the fact that only one operator performed pair slaughter.
Addressed on line 343: “The unbalanced sample size for this study, with the small number of co-slaughtered horses and only one operator co-slaughtering horses, is a potential limitation” and line 389: “The bias towards this breed type – and associated temperament – for co-slaughter could also have influenced behaviour towards the operator and propensity towards aggression or avoidance.”
L337-339 I think this is a fair point in the format your abattoir operated (with only one other horse in the kill pen and it being shot very soon after the first). Potentially caution would need to be taken with operationalising this to ensure that it is not interpreted as ‘horses aren’t distressed when seeing another horse killed’ so license is granted to have many animals present and/or long waits before the ‘witnessing animal’ is slaughtered.
Have added: ‘with a short time between shots and only two horses in the pen at any time,’ to clarify.
L371 Square brackets needed
Addressed.
Conclusion
377-379: I think more caution is needed here in your interpretation as you did not directly measure the handling status and familiarity of the animals with humans prior to their slaughter. Perhaps focus more on the paired vs individual aspect. Although I agree that the study results are important to take into account in countries where larger amounts of feral horses are slaughtered.
Addressed, line 404 amended to: “This study was the first of its kind to empirically analyse the impact that the presence of a conspecific can have on equine behaviour and welfare at slaughter.”

Reviewer 2 Report
1. What is the main question addressed by the research? The main question is whether having a companion horse mitigates stress at slaughter.
2. Do you consider the topic original or relevant in the field, and if
so, why? Yes, it is novel because there are fewer equine slaughter facilities than those for tradition meat producing animals
3. What does it add to the subject area compared with other published
material? It is a subject area that has received little attention.
4. What specific improvements could the authors consider regarding the
methodology? The methodology is fine. The authors are to be commended in using videos that had already been made so that more horses did not have to be slaughtered
5. Are the conclusions consistent with the evidence and arguments Yes
presented, and do they address the main question posed?
6. Are the references appropriate? Yes
24 don't you mean shooting of the first horse seldom elicited a startle response
Equine is an adjective! Use horse
38 and 39 remove % It was not that 12 % showed increased movement Perhaps 2 sentences. One with number and % of horses that were slaughtered in pairs and another with the % that exhibited behavior
66 omit However
72 United Kingdom
76 ensure the safety
77 Current knowledge does not provide evidence
Table 1 Didn't kicking with one hind leg count?
How did you rate asymmetric ears-one forward, the other back? What was scoring for shot effectiveness
148 what do you mean by crude association. If there was a crude association, were more tests performed? besides
"Paired- slaughtered horses moved more whilst in the kill pen (62%; 20/32, compared to 33%; 73/224 of individually slaughtered horses), although more individually slaughtered horses showed significant movement (8%; 17/224 compared to 3%; 1/32 of paired slaughtered horses). " Confusing Is there a difference between movement and significant movement?
158 How many horses were removed from kill pen ( i.e., used only to facilitate another horse's entry) This information is on line 262. Ten is th answer
181 definition of significant movement (#2 score) ?
Table 5 table needs to be better aligned
Don't understand Fig 1 what is difference between pair in pen and co-slaughter. Is one horse removed in pair in pen situation. This is explained earlier, but reader should be reminded.
256 Slips/falls more significant
279 more falls when unrestrained? for which group
Table 8 indicate direction of difference
330 indicates not evidences 337 fear response was hypothesized, but you didn't observe that did you?
329 mouth movements ( bite intention?)
334 What is social motivation probably should omit social
362 because not due to
3 70 reword slaughter; these horses are not accustomed
375 common not prolific
What was defined as affiliative behavior toward the operator?
However, being brought into the pen with a conspecific, which was then re-moved prior to slaughter, did not seem to be enough to mitigate stress behaviour. Did I miss this part of the results. There were 10 horses in that category
just a few re-wordings
Author Response
On behalf of my co-authors and myself, I would like to express our sincere gratitude to the reviewers for their useful feedback regarding our research article, entitled ‘Impact of social buffering and restraint on welfare indicators during UK commercial horse slaughter’. I have responded to the reviewers’ comments both online and, for your convenience, below, and attach a revised version of the article.
Reviewer 2:
24 don't you mean shooting of the first horse seldom elicited a startle response
Addressed.
Equine is an adjective! Use horse
have checked to ensure equine is not used where it shouldn’t be.
38 and 39 remove % It was not that 12 % showed increased movement Perhaps 2 sentences. One with number and % of horses that were slaughtered in pairs and another with the % that exhibited behavior
Addressed.
66 omit However
Addressed – ‘however’ removed on line 69.
72 United Kingdom
Have added ‘slaughterhouse personnel in the UK to line 77.
76 ensure the safety
Addressed.
77 Current knowledge does not provide evidence
Added ‘scientific knowledge’
Table 1 Didn't kicking with one hind leg count?
Addressed/amended.
How did you rate asymmetric ears-one forward, the other back?
This was rated as forwards, have amended table of descriptors to clarify this.
What was scoring for shot effectiveness?
This is clarified in Table 2: “signs of insensibility/consciousness: defined by: immediate collapse + lack of righting reflex + leg kicking or muscle spasms or rigidity.”
148 what do you mean by crude association. If there was a crude association, were more tests performed? besides
Crude association were assessed as part of the descriptive analysis - given the small number of observation in some categories, we joined some categories before conducting logistic regression (details are presented in table 3). Collinearity among predictor variables was checked before conducting multivariate analysis. Given that slaughter grouping and level of restraint, as well as level of restraint and time in pen had strong collinearity, results from the univariate logistic regressions were kept. We have edited the text to improve clarity (see line 165): “However, these horses were included in the individual slaughter category due to low numbers (n=10). Odds Ratios (OR) and 95% confidence interval (CI) were calculated as measure of strength of association. Collinearity was assessed between slaughter grouping and level of restraint and between level of restraint and time in pen. These three variables exhibited strong collinearity and therefore univariate models were kept.”
"Paired- slaughtered horses moved more whilst in the kill pen (62%; 20/32, compared to 33%; 73/224 of individually slaughtered horses), although more individually slaughtered horses showed significant movement (8%; 17/224 compared to 3%; 1/32 of paired slaughtered horses). " Confusing Is there a difference between movement and significant movement?
Addressed, and all tables altered to ensure consistency in terminology.
158 How many horses were removed from kill pen ( i.e., used only to facilitate another horse's entry) This information is on line 262. Ten is th answer
Addressed.
181 definition of significant movement (#2 score) ?
Addressed as per reply to reviewer one.
Table 5 table needs to be better aligned
Edited to be better aligned.
Don't understand Fig 1 what is difference between pair in pen and co-slaughter. Is one horse removed in pair in pen situation. This is explained earlier, but reader should be reminded.
We have given more details in the figure caption to improve clarity. Now reads: “Time spent in kill pen prior to slaughter and (a) whether the animal was in the pen individually; brought in with a conspecific and co-slaughtered; or brought in with a conspecific, which is then removed leaving the animal alone for shooting, (b) type of grouping in pen and shot order, and (c) restraint method, showing differences in time for all these factors (slaughter grouping, shot order and level of restraint).”
256 Slips/falls more significant
Have added ‘more’ to line 270 to read: “Slips/falls was found to be more significant”
279 more falls when unrestrained? for which group
Just for unrestrained group- have added ‘When restraint was analysed as a separate factor to slaughter grouping,’ to clarify this was analysed separately.
Table 8 indicate direction of difference
The direction is indicated by whether the Odds Ratio is positive or negative and which is the Reference.
330 indicates not evidences
Addressed/amended to “suggests” to avoid duplication of word “indicates” elsewhere in sentence.
337 fear response was hypothesized, but you didn't observe that did you?
Changed to ‘less frequently than expected’
329 mouth movements ( bite intention?)
This was generally interpreted as possible vocalisation due to occurrences at a distance from the operator or conspecifics.
334 What is social motivation probably should omit social
Addressed
362 because not due to
Addressed
370 reword slaughter; these horses are not accustomed
Re-worded for clarity to ‘with these horses not accustomed to, or familiar with, being haltered’
375 common not prolific
Addressed
What was defined as affiliative behavior toward the operator?
Have added this definition to table 1.
However, being brought into the pen with a conspecific, which was then re-moved prior to slaughter, did not seem to be enough to mitigate stress behaviour. Did I miss this part of the results. There were 10 horses in that category
Table 8 presents these results, which show that, whilst being brought in with a conspecific (which was then removed prior to slaughter) appeared to reduce movement and reactivity, it did not significantly affect behaviour towards the operator, ear posture or ease of entry, have added this on line 281.
